# Medical School Education on Myalgic Encephalomyelitis

**DOI:** 10.3390/medicina57060542

**Published:** 2021-05-28

**Authors:** Nina Muirhead, John Muirhead, Grace Lavery, Ben Marsh

**Affiliations:** 1Buckinghamshire Healthcare NHS Trust, Amersham Hospital, Whielden Street, Amersham HP7 0JD, UK; 2Boston Consultants Ltd., Solihull, West Midlands B93 8PG, UK; john.muirhead@btinternet.com; 3School of Medicine, Cardiff University Medical School, Neuadd Meirionnydd, Cardiff CF14 4YS, UK; LaveryGE@Cardiff.ac.uk; 4University Hospitals Plymouth NHS Trust, Derriford Hospital, Derriford Road, Plymouth, Devon PL6 8DH, UK; bmarsh@doctors.org.uk

**Keywords:** ME/CFS, education, medical school, teaching, patient safety, NICE Guidelines, Health Act 1983, General Medical Council, GMC, Medical Schools Council, MSC, long Covid

## Abstract

*Background and objectives:* Myalgic Encephalomyelitis/Chronic Fatigue Syndrome (ME/CFS) is a complex multi-system disease with a significant impact on the quality of life of patients and their families, yet the majority of ME/CFS patients go unrecognised or undiagnosed. For two decades, the medical education establishment in the UK has been challenged to remedy these failings, but little has changed. Meanwhile, there has been an exponential increase in biomedical research and an international paradigm shift in the literature, which defines ME/CFS as a multisystem disease, replacing the psychogenic narrative. This study was designed to explore the current UK medical school education on ME/CFS and to identify challenges and opportunities relating to future ME/CFS medical education. *Materials and methods*: A questionnaire, developed under the guidance of the Medical Schools Council, was sent to all 34 UK medical schools to collect data for the academic year 2018–2019. *Results:* Responses were provided by 22 out of a total of 34 medical schools (65%); of these 13/22 (59%) taught ME/CFS, and teaching was led by lecturers from ten medical specialties. Teaching delivery was usually by lecture; discussion, case studies and e-learning were also used. Questions on ME/CFS were included by seven schools in their examinations and three schools reported likely clinical exposure to ME/CFS patients. Two-thirds of respondents were interested in receiving further teaching aids in ME/CFS. None of the schools shared details of their teaching syllabus, so it was not possible to ascertain what the students were being taught. *Conclusions:* This exploratory study reveals inadequacies in medical school teaching on ME/CFS. Many medical schools (64% of respondents) acknowledge the need to update ME/CFS education by expressing an appetite for further educational materials. The General Medical Council (GMC) and Medical Schools Council (MSC) are called upon to use their considerable influence to bring about the appropriate changes to medical school curricula so future doctors can recognise, diagnose and treat ME/CFS. The GMC is urged to consider creating a registered specialty encompassing ME/CFS, post-viral fatigue and long Covid.

## 1. Introduction

Myalgic Encephalomyelitis/Chronic Fatigue Syndrome (ME/CFS) affects around 250,000 patients in the United Kingdom (UK); it is twice as common as other diseases that feature in undergraduate curricula, such as Human Immunodeficiency Virus (HIV) and Multiple Sclerosis (MS). In a recent survey of 4038 ME/CFS patients, 62% stated they are not confident their General Practitioner (GP) understands the condition, and 18% of patients wait longer than six years for a diagnosis [1]. The impact of this disease on patients’ wellbeing [2] and quality of life is significant compared with other diseases [3]; yet, between a third and half of GPs lack confidence in acknowledging, diagnosing and managing ME/CFS [4], and the disease is often incorrectly dismissed as psychosomatic [5]. Davenport et al. note that up to 90% of patients are undercounted, undiagnosed and under-treated [6].

ME/CFS is a complex, multi-system disease, diagnosed on a history of significant fatigue impairing function, post exertional malaise, unrefreshing sleep, orthostatic intolerance and/or cognitive impairment [7]. Unlike any other illness and disease, advice to exercise is contraindicated. Exercise in ME/CFS has been shown to result in symptom exacerbation, deterioration of cellular bioenergetics and increased disability. A growing number of recent studies demonstrate abnormalities in cognition, brain changes on spectroscopy scans, lower metabolic energy generation and altered immune system response as well as neuroinflammation following repeated exercise [8].

In 1998, The Chief Medical Officer (CMO) of the UK appointed an Independent Working Group (IWG) to investigate divergent clinical views of ME/CFS and dissatisfaction among patients and patient support groups about the paucity of medical services to deal with this disease [9]. The IWG, which was first to acknowledge the importance of the patient voice, published their report in 2002, recommending that: “improvements are needed in the education and training of doctors, nurses and healthcare professionals, especially in primary care; ME/CFS should be considered as a differential diagnosis and GPs and medical specialists should be able to provide basic guidance after diagnosing this condition”.

Given that, 20 years later, patients and patient support groups continue to be dissatisfied with the healthcare community’s response to ME/CFS, this study was undertaken to establish the extent to which medical schools are covering this subject in their curricula and, if possible, why healthcare professionals still seemingly struggle to understand ME/CFS or, in some cases, deny the existence of this disease other than as a mental health condition [4,5].

In November 2020, the UK’s National Institute for Health and Care Excellence (NICE) issued new draft guidelines [10] on ME/CFS. These acknowledge that ME/CFS is a chronic multi-system medical condition with distinct clinical diagnostic criteria. Echoing the 2002 CFS/ME IWG report, NICE calls for significant improvements in the education of healthcare professionals with greater emphasis on the delivery of evidence-based training to represent current knowledge and the experiences of people with ME/CFS.

The fact that NICE in 2020 makes virtually the same recommendations as the IWG in 2002 demonstrates serious failures in medical education in ME/CFS over the past almost 20 years. European ME/CFS experts have expressed serious concerns about knowledge and understanding among primary care physicians, and survey responses demonstrated that 91% strongly agreed there should be more teaching about ME/CFS in undergraduate medical curricula [11].

This study establishes a baseline of how and to what extent the subject of ME/CFS is being taught in UK medical schools and reveals an exciting opportunity to research the pedagogy surrounding a paradigm shift in a disease narrative. Knowledge of this complex multi-system disease has been hindered by a failure to “move on”. We can no longer describe ME/CFS as a figment of patients’ unhelpful beliefs, and the burden of ME/CFS in the wake of COVID is an opportunity to learn [12]. Improved medical education on the topic of ME/CFS is urgently required to improve patient safety.

## 2. Materials and Methods

Approval of the UK Medical Schools Council was obtained before this study was undertaken. The study was advocated by Forward ME, Cardiff University and the CFS/ME research collaborative (CMRC).

A questionnaire comprising ten questions was developed to ascertain the extent of current teaching ME/CFS in all UK medical schools. The Medical Schools Council circulated a request to all 34 schools in the UK in October 2018, this invited schools to participate voluntarily in the study and providing them with a link to the online questionnaire (using Survey Monkey). E-mail reminders were sent in February and March 2019. Not all schools responded, and some responded anonymously.

## 3. Results

Out of a total of 34 schools, 22 responded (65%), of which 13 schools taught ME/CFS in their syllabuses (59%), leaving nine schools (41%) that did not.

### 3.1. Teaching Methodology

As Figure 1 shows, nine schools out of 13 (69%) taught by lecture, five used discussion and/or case study methods and some stated that the “Unrest” video [13] had been shown and formed a part of their discussions. E-Learning, tutorial and handouts were less frequently used. Some schools use more than one method; a single method was used by seven schools, two methods were used by five schools, and four methods were used by one school.

### 3.2. Teaching Duration

The nine medical schools who responded that they do not teach this subject are included here as zero hours (h). Eight schools devoted between 1 and 2 h to teaching ME/CFS; two schools devoted more than 3 h while one school devoted less than 1 h to the subject; one school was unable to quantify teaching duration. See Figure 2.

### 3.3. Part of Curriculum Covering ME/CFS Teaching

On average, ME/CFS was taught within two parts of the curriculum, described here as medical disciplines. Figure 3 shows that ME/CFS across the 13 schools was taught by at least six different medical disciplines. The most common was General Practice (*n* = 5); followed by Chronic Disease, Neurology and Psychiatry (all *n* = 4), Rheumatology (*n* = 3) and Paediatrics (*n* = 1), details were not provided for ‘other’.

### 3.4. Medical Specialists Leading Teaching of ME/CFS

Various specialists provided the core of teaching for ME/CFS, as shown in Figure 4. Some supplied more than one specialist to teach the subject, seven schools referred to professors or senior teaching fellows without stating their area of expertise, and ten different specialists were listed. Psychiatrists (*n* = 5) and general practitioners (*n* = 4) were the dominant specialists in ME/CFS teaching, followed by rheumatologists (*n* = 3) and neurologists, general medicine and public health specialists (*n* = 2). ME/CFS teaching was also delivered by behavioural scientists, infectious disease specialists, ophthalmologists and clinical communicators (each *n* = 1).

### 3.5. Clinical Contact with ME/CFS Patients

Only three schools out of 13 (23%) responded affirmatively to the inclusion of contact with ME/CFS patients as part of their curriculum.

### 3.6. Examination Practices

The following results relate to all 22 respondents irrespective of whether they taught ME/CFS in their curriculum. Seven schools out of 22 (32%) stated that they set questions on the subject in their examinations.

### 3.7. Interest in Further Teaching Aids

Fourteen schools out of 22 (64%) stated that they were interested in receiving further teaching aids on the subject of ME/CFS. Of the nine schools that do not teach ME/CFS, seven schools (78%) said they were interested in receiving further teaching aids or materials.

The most common teaching aid of interest was educational videos of 20–30 min duration, followed by e-learning module of 30–60 min duration or lecture with patient volunteers of 30–60 min duration. Each of these options was preferred by five schools (note: not necessarily by the same five schools). Three schools showed an interest in a lecture of 30–60 min duration. A total of 27 options were chosen by 14 schools, an average of almost two per school; see Figure 5.

## 4. Discussion

### 4.1. Potential Bias

The lack of response from some medical schools could bias the results of this study to over- or under-estimate the current teaching. Given the high level of undiagnosed sufferers with ME/CFS, the low level of confidence among GPs to be able to diagnose this disease [14] and the absence of patient satisfaction in the medical profession [1], it is plausible that 41% of medical schools do not teach ME/CFS. From the data gathered, the actual number of medical schools that cover this topic could lie be between 38% and 73%. This study would be more accurate with more respondents; however, the 64% overall response rate across the UK is greater than the 54% response rate in published research on ME/CFS teaching in medical schools in the United States [15] and other similar UK medical school surveys on ageing [16], neuroanatomy [17] and frailty [18], which had 19/30 (63%), 24/34 (70%) and 25/34 (74%) medical school responses, respectively. Another way of verifying if medical schools have timetabled teaching on ME/CFS would be to explore the existing data, already carried out for the 2014/15 academic year, with 47,258 timetabled teaching events in 25 UK medical schools [19].

### 4.2. Teaching Time and Methodology

Of the 59% that do cover ME/CFS, teaching duration is usually about one hour, it is not always examinable, and few augment their teaching with exposure to patients with the disease. One to two hours of teaching seems to be very low for a common chronic disease. A typical UK medical student receives 3960 timetabled hours of teaching during their five-year course [19]. Other research showed that the mean amount of core neuroanatomy teaching was 29.3 h [17], and the median time spent on teaching ageing and geriatric medicine was 55.5 h [16].

Another limitation of the study was the lack of information provided on what is being taught, which leaves us unable to comment on the quality and content of the teaching. Indeed, there is a risk that teaching could misrepresent the illness, or categorise it as psychosomatic. Therefore, teaching outdated content could be far worse than not teaching undergraduates about ME/CFS.

Little seems to have changed since a study in 2008 [20], which revealed “Family physicians obtain information about [ME/CFS] from their nonprofessional world which they incorporate into their professional realm”. A more recent analysis of ME/CFS teaching in one UK medical school [21] concluded that “Students acquired their knowledge and attitudes largely from informal sources and expressed difficulty understanding [ME/CFS] within a traditional biomedical framework”, which is further evidence that education improvements proposed by IWG in 2002 have not been implemented.

### 4.3. Medical School Curriculum

Respondents were invited to send their syllabuses to enable a more detailed analysis of what is being taught about ME/CFS in their respective schools. A similar study undertaken in the United States revealed only 5.6% of medical schools were judged to deliver sufficient clinical, curricula and research on ME/CFS [15]. However, as no syllabus details were provided by any of the respondents and no explanations given, it was not possible to throw any light on why many healthcare professionals in the UK still struggle to recognise this disease, be able to diagnose it or agree upon suitable management or treatments.

The wide spectrum of medical specialists that are involved in teaching ME/CFS, as revealed by this study, could explain why healthcare professionals remain confused. Whilst ME/CFS is a complex, multi-system disease that will continue to attract a variety of theories at a research level, there is no apparent reason why undergraduate medical students cannot be taught how to recognise and diagnose this disease and be able to make recommendations on its management. The over-riding priority in undergraduate teaching is to improve attitudes towards patients and acknowledgement of genuineness of the patient experience and validity of the disease. Some treatment approaches should have no place in undergraduate teaching; especially those that are shown to cause patient harm, delayed diagnosis and unsafe advice to exercise, as well as outdated assumptions such that dysfunctional beliefs, behaviours or even personality traits that are responsible for causing or perpetuating this illness [22].

### 4.4. Medical Education Challenges

Despite almost twenty years of stagnation, there is now a substantial need for ME/CFS medical education to “move on”, and the 64% interest in further teaching aids is encouraging. It is proposed that the paradigm shift in international understanding of this condition [12], along with a lack of specialists, is an opportunity for medical educators to develop new teaching materials for medical schools to use in a flipped classroom model. Such materials, updated to reflect the latest biomedical science developments and patient perspectives, would transform what is taught. Regarding diagnosis and management of ME/CFS, a recently developed online module [23] has shown that such measures significantly increase confidence in recognising diagnostic criteria. Teaching could be augmented with patient videos [24], webinars and podcast interviews to convey both the complexity and patient experience of this disease. Over the last twenty years, there have been huge strides in online communication, patient support groups on social media and the emergence of the ‘patient expert’. The patient voice and perspective are also becoming central to medical education; there is an opportunity for medical schools to work with networks of patients and family members, who have an existing wealth of knowledge, to assist in augmenting future medical education.

This proposal would, furthermore, comply with the new NICE draft guidelines [10], which call for improvements in evidence-based education and training of healthcare professionals and better acknowledgment of the patient experience.

A much broader question arising from this study is why ten different specialties were involved in teaching this subject. Medical education is already moving away from specialty silos, but the secondary care system remains poorly equipped to manage the needs of patients with complex multisystem disease. ME/CFS patients are often cycled through multiple secondary care specialists, with the potential for each hospital visit to exacerbate their symptoms. Apart from the obvious economic drain on resources, the effect on patients and their families may explain why so many patients disconnect from the healthcare system. Clinicians with a knowledge and understanding of ME/CFS could reduce harm, save resources, improve patient care, limit delays to diagnosis and remove misplaced advice to exercise.

Based upon the findings in this study, the UK General Medical Council (GMC), which has statutory responsibilities under the Health Act 1983 for medical education curricula and standards, and the Medical Schools Council (MSC), which represents medical schools in various areas of common interest, are called upon to use their considerable influence to bring about changes in medical schools’ undergraduate and postgraduate curricula so that doctors of the future are more capable of recognising, diagnosing and treating ME/CFS. Additionally, the GMC may also wish to consider recognition of ME/CFS as a specialty, which could also encompass post-viral fatigue and the growing subset of long Covid patients who present with ME/CFS.

The authors are not aware of any earlier study into the extent and nature of ME/CFS medical education across UK medical schools. This study therefore provides a baseline as to where UK medical education currently stands in relation to quantity, although it is difficult to comment on the content or quality of teaching in this subject.

### 4.5. Further Research

While this study is merely exploratory, it provides evidence that further research is required into what is being taught, whether this is evidence based, how it is assessed and how this might affect student knowledge and attitudes towards ME/CFS patients and their families.

Medical students from a variety of UK medical schools could be surveyed on their knowledge and perception of ME/CFS, what they have learned during medical school and how they think the undergraduate curriculum might adapt to improve ME/CFS education. The paradigm shift in ME/CFS literature and guidelines provides new opportunities for medical education research, which could be designed to measure changes in knowledge and/or attitudes and beliefs following updated teaching interventions. The lack of disease recognition and delays to ME/CFS diagnosis are not only a challenge in the UK, but also worldwide; this study and its findings are relevant to international colleagues researching ME/CFS education in other countries.

## 5. Conclusions

UK medical education in ME/CFS is currently inadequate and appears not to have progressed over the past two decades. Of the medical schools responding, 41% do not teach the subject at all. Data on the 59% of the medical schools that do cover ME/CFS show that education is delivered by multiple medical specialists, mostly by lectures of one-hour duration, which is not always examinable and often takes place without any exposure to patients with the disease.

Differences in beliefs of medical specialists concerning the pathogenesis of ME/CFS need to be set aside in the interest of improving the clarity of what is taught at undergraduate level with renewed focus on diagnosis and management, acknowledging and believing the patient and their families, as well as treating patients with care, empathy and compassion.

Many medical schools (64% of respondents) acknowledge the need to improve education and training of healthcare professionals by expressing a strong appetite for more teaching aids and materials that convey the complexity of this disease. The GMC and MSC are encouraged to use their considerable influence to bring about change in medical schools’ curricula in ME/CFS.

## Figures and Tables

**Figure 1 medicina-57-00542-f001:**
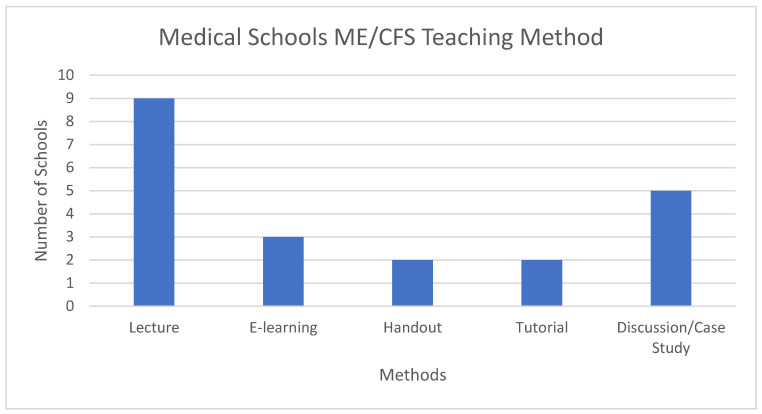
Teaching Methodology. ME/CFS, Myalgic Encephalomyelitis/Chronic Fatigue Syndrome.

**Figure 2 medicina-57-00542-f002:**
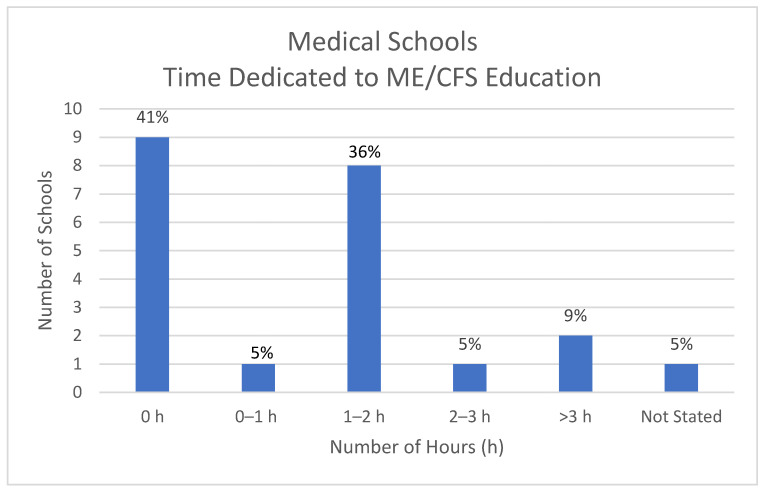
Teaching Duration.

**Figure 3 medicina-57-00542-f003:**
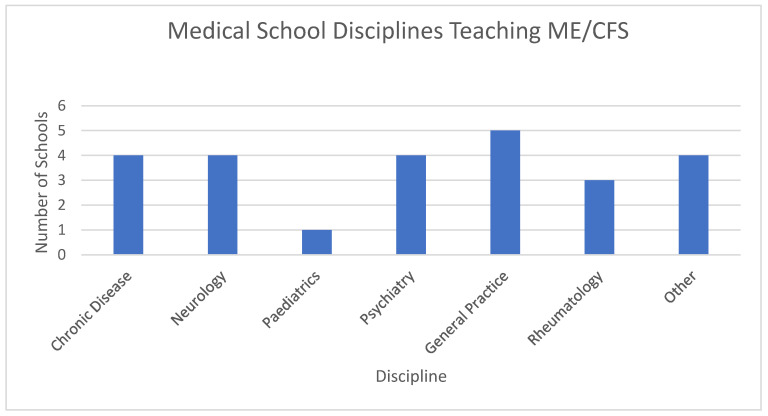
Disciplines Providing ME/CFS Teaching.

**Figure 4 medicina-57-00542-f004:**
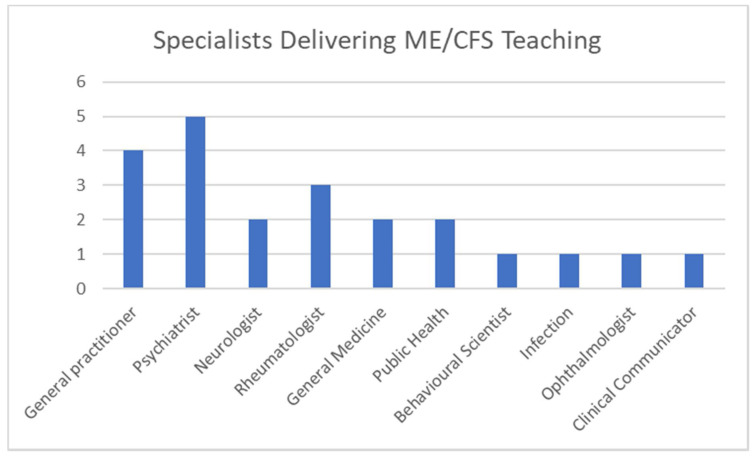
Medical Specialists Leading Teaching of ME/CFS.

**Figure 5 medicina-57-00542-f005:**
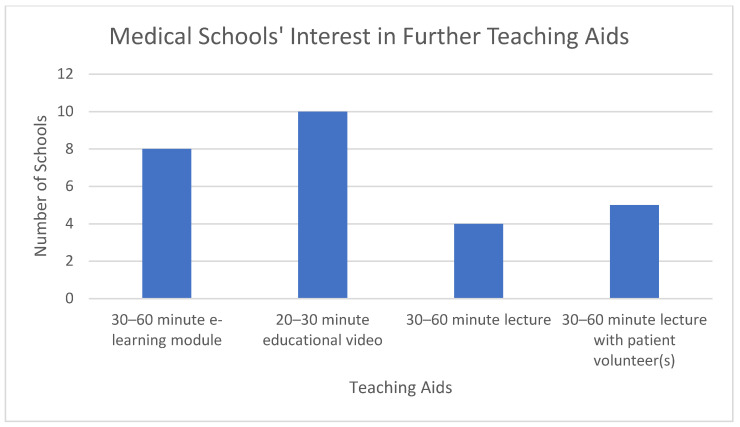
Interest in Further Teaching Aids.

## Data Availability

The anonymous data presented in this study are available on request from the corresponding author.

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
