# Peer review of "Medical School Education on Myalgic Encephalomyelitis"

_medicina, 2021, doi:10.3390/medicina57060542_

Round 1

Reviewer 1 Report

In overall, the study explore a very significant subject in education in biomedical field in general, namely the ways and efficacy of teaching. In the case of ME/CFS, it is in of particular importance, because it could be assumed  that it is under diagnosed due to the lack of knowledge in some of clinicians. The nature of study is mainly exploratory, without application of more sophisticated statistical models. However, results of this study could serve in in setting directions for future research. I have found seven really minor issues and one major that need to be fixed.

Major issue: The nature of the current study is exploratory, what should be underlined in study limitations. Without proper testing of hypotheses using statistical analysis it is not proper to draw strong conclusions from the current study. Therefore, both in abstract and in the main part of text, some conclusions should  be reworded. For instance conclusion written in the abstract “UK medical school teaching in ME/CFS is shown to be inadequate” cannot be directly drawn from Your study/based on result of analysis. The same seems to apply to conclusion “This study proposes that educators could sharpen focus and deliver a more coherent evidence-based message to enable healthcare professionals in the recognition, diagnosis and treatment of this disease.” This study cannot directly serve as a justification for this changes. However, based on this study, You as Authors can propose a direction of further studies. Effects of evidence-based courses on ME/CFS (on some indicators that You consider as “objective improvement” in healthcare system regarding ME/CFS patients) could be examined in the further (RCT type?) studies.

List of minor issues:

  1. Please explain abbreviations for GMC and MSC in the abstract
  2. Introduction: „CFS/ME should be considered as a differential diagnosis” could You elaborate on this? Differential diagnosis „is a very interesting part, especially if we consider a ME/CFS diagnosis as a negative diagnosis i.e. „patients has ME/CFS if he/she is not suffering from something else that might explain symptoms”
  3. Introduction „healthcare professionals still struggle to understand ME/CFS or, in some cases, to deny the existence of this disease other than as a mental health condition.” Indeed, that could be a huge issue, could You add a citation to this information?
  4. Results: Could You add percentages on the top of the columns? It applies to all figures
  5. Results: “7 schools referred to professors or senior teaching fellows without stating their area of expertise” please start this sentence with „Seven”
  6. Discussion: “The lack of response from some medical schools could bias the results of this study to overestimate the current teaching.” I would speculate the same, but do You have any evidence (reference) why it would lead to OVERestimation and not to UNDERestimation?
  7. Discussion “It is proposed that the paradigm shift in international understanding of this condition is an opportunity for medical educators to develop clear teaching materials for medical schools to use in a flipped classroom model.” Are You referring to Thomas S. Kuhn definition of paradigm shift? If so, what are the basis of this statement? Or maybe You are referring here just to an increase in number of scientific publications on ME/CFS during the last years?

Author Response

The authors would like to thank the reviewer for the time and consideration given to our paper. The points raised were very constructive in providing feedback and we have agreed with all of them and hope that the amendments are satisfactory.

Major issue:

We agree that this is an exploratory study and that results of this exploratory study could serve in in setting directions for future research.

We have re-worded the abstract “UK medical school teaching in ME/CFS is shown to be inadequate” as we agree this cannot be directly drawn from this data.

The statement “This study proposes that educators could sharpen focus and deliver a more coherent evidence-based message to enable healthcare professionals in the recognition, diagnosis and treatment of this disease.” Has also been amended.

We have chosen further re-wording to ensure that we have represented the study more accurately in its limitations and exploratory nature.

List of minor issues:

GMC and MSC are now written out in full in the abstract.

Introduction: „CFS/ME should be considered as a differential diagnosis” this is a quote from the report.

Introduction: Two citations have been added in support of the comment “healthcare professionals still struggle to understand ME/CFS or, in some cases, to deny the existence of this disease other than as a mental health condition”

Results: Percentages have not been added to the columns of all graphs as several medical schools answered more than one question, not all medical schools answered every question and therefore the numbers are not out of 100%, apart from hours spent teaching where the percentages have now been added.

Results: Numbers ten and under have been expressed as words instead of figures.

Discussion: “The lack of response from some medical schools could bias the results of this study to overestimate the current teaching.” – Discussion relating to this has been expanded and referenced to address this point more accurately.

Discussion “It is proposed that the paradigm shift in international understanding of this condition is an opportunity for medical educators to develop clear teaching materials for medical schools to use in a flipped classroom model.” Yes, this is Thomas S. Kuhn definition of paradigm shift. Paradigm Lost: Lessons for Long COVID - The Science Bit [1]

Once again, we hope our changes are acceptable and have found the feedback incredibly useful. The acknowledgements section has been updated to recognise the role of peer review in improving the paper, thank you.

[1] “Paradigm Lost: Lessons For Long COVID-19 From A Changing Approach To Chronic Fatigue Syndrome, " Health Affairs Blog, May 18, 2021.DOI: 10.1377/hblog20210514.425704

Reviewer 2 Report

A well presented paper on an important subject, i.e. that of education.

Minor comments for review:

Some places would benefit from references, e.g. when describing ME/CFS as a complex multi-system disease.., it is probably IOM reference. Unrest video reference and low confidence of GPs are another instance when a reference would be appropriate.

Fig 1: would be useful to indicate values for "more than one method"

3.2. Please indicate how many schools provide "zero hours" of teaching in the subject.

Fig 3. Other disciplines. would be useful to have these listed, if information is available, e.g. as a foot note to the Fig.

Discussion: Bias could be expanded on, the authors mention that 64% response rate is significant and sufficient to draw valid conclusions. Could they expand on which criteria they used to come to this conclusion? I would expect this information to be easily available to the Universitities.Have the authors  tried to obtain the information using other ways ?

If lack of response implies no teaching on the subject, then what proportion of school offered any or no teaching? This could be reported in the text (and also the opposite assumption, all non-respondents did teach), so  you have a range of results that will include.

An important limitation is that the content of the teaching is not reported. The question remain on the quality of the teaching. Are the right concepts being taught? If there is a risk teaching will not reflect the current values and understanding of the disease, then there is a risk here which could be as high or higher than the risk of no teaching. This is something the authors could mention and discuss. Also the number of hours of teaching in the subject seems very low, do the authors think this should be emphasised?

4.4/. Long covid patients (please add "who present with ME/CFS"

Author Response

Thank you so much for the time you have taken to review our paper and for your very constructive and helpful feedback.

References have been added for the complex disease definition, “Unrest” and low GP confidence.

Fig 1: values have been added in the results description for "more than one method"

3.2. Has been amended to indicate how many schools provide "zero hours" of teaching in the subject.

Fig 3: Other disciplines have not been detailed but other specialties were listed and the graph has been expanded to show these.

The 64% response was compared against other similar newly referenced research (ageing, frailty, neuroanatomy) and other ways which may prove helpful in designing future studies (e.g. looking at the raw data from medical school timetables).

Discussion:

If all those who did not respond do teach the subject, or vice versa this gives a range of results of those who do teach the subject as between 38% and 73%, we have added this information.

The lack of information gathered on content of teaching is discussed. We agree that if there is a risk teaching will not reflect the current values and understanding of the disease, then there is a risk here which could be as high or higher than the risk of no teaching.

We agree that the number of hours of teaching in the subject seems very low and we have tried to emphasise this through comparison with existing literature.

Long covid patients – now reads more accurately as the subset "who present with ME/CFS"

Thank you so much for the peer review, and the time you have dedicated towards improving the accuracy of the information and relevant discussion for this publication. We have acknowledged your very valuable contributions in the acknowledgements section.

Reviewer 3 Report

This is an important review revealing the inadequate efficacy of medical student education on  the nature and management of ME/CFS patients in the UK system despite teaching on the condition occuring at a significant  proportion of the medical schools in contrast to many countries. The results are disappointing for me personally, as a member of a ME/CFS family and a biomedical researcher who has been determining  the biological basis of the disease.

The collected information highlights an important conclusion that different disciplines carry out the education in different schools. This identifies a huge deficit in the teaching programmes as ME/CFS is a complex disease that affect many systems and patients have been very poorly served by the 'silo system' where the disease is viewed differently by specialists in different disciplines. As the report here concludes differences need to be set aside and there needs to be a broad evidence based approach to provide 'greater clarity' for undergraduates. My own experience is that students are very receptive to that - the most common question i am asked is "what do I do when I meet an ME/CFS patient". My response is 'first affirm they are ill even if you do not understand their disease'. The recommendations in the report are excellent but I  would  go one step further.  Not only should the teaching  involve 'patients' but also families. Alone it should be acknowledged patients are very vulnerable with their symptoms ( memory deficiencies, brain fog, together with their serious fatigue) and there can easily be an imbalance of power. There is  a significant effect on families and it is a critical part of managing the patient's illness  that should be part of the in depth medical education, in my view.

The teaching methodolgy used by the schools documented here  is interesting but the number of schools using multiple modailites is disppointingly low. The "Unrest" video is an excellent resource but even Jennifer Brea's TED talk that is much shorter is an effective  stark description of how an individual is affected. In my view face to face teaching, backed with tutorial discussion and accessing families with patients are all important facets needed. Critical to my family's interaction with the medical system was that it was  not aligned with the needs of the patient particularly when they are in the acute phase. Patients waiting in crowded waiting rooms to see their practitioner, patients needing  extended sleep impossible with  hospital routines when hospitalised, are examples of this. The fact that a patient can get multiple and different diagnoses by cycling through the 'silo' specialities encourages patients to disconnect from what is perceived to be an unhelpful medical system.

The dedication of only 1-2 h of medical education devoted to ME/CFS, given  the incidence of the disease, is identified by this report as woefully inadequate   and it is hoped with the advent of post viral Long COVID will 'kick start ' medical schools to start taking ME/CFS  much more seriously.

Figure 3 illustrates the silo approach in the current teaching - hopefully within a chronic disease umbrella the other disciplines can all give their part in a co ordinated way but with the same 'message'. My own experience is teaching is in an 'unexplained disease' module organised by general practice that has merit but without interest and even discouragement to delve into new evidenced based biomedial research  on what has  been recently discovered  about ME/CFS. In my experience in my country, sadly,  the specialists I have come across in the areas  illustrated in Figure 4 (ie those teaching)- do not have any where near an in depth understanding of ME/CFS ( how can they teach about the disease when their knowledge and understanding is so lacking?). One shining light is  a community  general practitioner  who has seen about 5000 ME/CFS patients, and her knowedge started while doing medical training when the Royal Free Hospital outbreak occurred in 1955 in London (from which the name ME originated).

The discussion and conclusions from this simple straightforward collection of data about the current medical education is very sound. An absolutely critical factor that has held up approriate teaching is the differences in beliefs in the aetiology and pathogenesis of ME/CFS. But in the last 5 years a wealth of new research identifying  the comprehensive physiological changes occurring in ME/CFS patients provides a firm basis for a new wave of teaching. It is time to put aside the differences in strongly held opinions, teach the evidence based new reasearch and involve familiers who have  a wealth of information on how the disease affects the patient and their family. The recent advent of self identifying social media groups of patients  advocating for themselves and educating each other and the public is a constructive addition to help drive this transformation in medical education that is so desparately needed by patients world wide.

I commend the authors for this report that reveals the inadequacies of current medical education in the UK on ME/CFS that I am sure is mirrored world wide and hope it provides a stimulus for medical schools to do better and to embrace the new understanding of the disease so the  future emerging health practitioners will be able to provide better care of patients with this difficult ongoing and debilitsating illness.

Author Response

Thank you for taking the time to review our work, the feedback was insightful, and we have expanded our discussion to include some of the key aspects identified as pertinent to this topic.

These include the many systems and symptoms associated with this disease and the way in which patients have been very poorly served by the 'silo system' where the disease is viewed differently by specialists in different disciplines. Figure 3 and 4 illustrate the silo approach in the current teaching and we have expanded on comments about this in both the abstract and discussion, also detailing the ‘other’ specialties who deliver teaching on this subject – there were ten different ones.

Our discussion has evolved to incorporate the need to acknowledge the patients and their families in medical education.

We agreed with the perspective that the dedication of only 1-2 h of medical education devoted to ME/CFS, given the incidence of the disease and have drawn readers towards the idea that there may be an increase of ME/CFS associated with post viral Long COVID.

We agree with the positive observation of new opportunities that have arisen including the advent of self identifying social media groups of patients advocating for themselves and educating each other and the public and have included this.

We thank you for supporting our publication and have acknowledged this formally in the acknowledgement section. We are greatly appreciative of the constructive contribution that your review has made to our paper and its discussion points.

This manuscript is a resubmission of an earlier submission. The following is a list of the peer review reports and author responses from that submission.